# Cell Profiling of Acute Kidney Injury to Chronic Kidney Disease Reveals Novel Oxidative Stress Characteristics in the Failed Repair of Proximal Tubule Cells

**DOI:** 10.3390/ijms241411617

**Published:** 2023-07-18

**Authors:** Zhixiang Yu, Ying Zhou, Yuzhan Zhang, Xiaoxuan Ning, Tian Li, Lei Wei, Yingxue Wang, Xiao Bai, Shiren Sun

**Affiliations:** 1Department of Nephrology, Xijing Hospital, Fourth Military Medical University, Xi’an 710032, China; 2Department of Geriatrics, Xijing Hospital, Fourth Military Medical University, Xi’an 710032, China; 3School of Basic Medicine, Fourth Military Medical University, Xi’an 710032, China; 4National Local Joint Engineering Research Center for Precision Surgery & Regenerative Medicine, Shaanxi Provincial Center for Regenerative Medicine and Surgical Engineering, Center for Regenerative and Reconstructive Medicine, Med-X Institute, First Affiliated Hospital of Xi’an Jiaotong University, 124, 76 West Yanta Road, Xi’an 710061, China; ywan3026@uni.sydney.edu.au

**Keywords:** acute kidney injury, chronic kidney disease, failed repair of PT cells, oxidative stress, intercellular communication, single-nucleus RNA sequencing

## Abstract

Chronic kidney disease (CKD) is a major public health issue around the world. A significant number of CKD patients originates from acute kidney injury (AKI) patients, namely “AKI–CKD”. CKD is significantly related to the consequences of AKI. Damaged renal proximal tubular (PT) cell repair has been widely confirmed to indicate the renal prognosis of AKI. Oxidative stress is a key damage-associated factor and plays a significant role throughout the development of AKI and CKD. However, the relationships between AKI–CKD progression and oxidative stress are not totally clear and the underlying mechanisms in “AKI–CKD” remain indistinct. In this research, we constructed unilateral ischemia–reperfusion injury (UIRI)-model mice and performed single-nucleus RNA sequencing (snRNA-seq) of the kidney samples from UIRI and sham mice. We obtained our snRNA-seq data and validated the findings based on the joint analysis of public databases, as well as a series of fundamental experiments. Proximal tubular cells associated with failed repair express more complete senescence and oxidative stress characteristics compared to other subgroups. Furthermore, oxidative stress-related transcription factors, including Stat3 and Dnmt3a, are significantly more active under the circumstance of failed repair. What is more, we identified abnormally active intercellular communication between PT cells associated with failed repair and macrophages through the APP–CD74 pathway. More notably, we observed that the significantly increased expression of CD74 in hypoxia-treated TECs (tubular epithelial cells) was dependent on adjacently infiltrated macrophages, which was essential for the further deterioration of failed repair in PT cells. This research provides a novel understanding of the process of AKI to CKD progression, and the oxidative stress-related characteristics that we identified might represent a potentially novel therapeutic strategy against AKI.

## 1. Introduction 

Acute kidney injury (AKI) is a common clinical syndrome that not only affects the short-term prognosis of patients but also induces chronic kidney disease (CKD), placing a heavy burden on individuals, families, and society [1,2]. However, the exact mechanisms of the transition from AKI to CKD remain unclear. Proximal tubular (PT) cells, the sizeable predominant proportion of renal-intrinsic cell types in the mammalian kidney [3], are sensitive to ischemia and more likely to be injured. After AKI, PT cells begin a series of repair processes, including dedifferentiation, migration, and proliferation, to recover normal kidney function and structure [4,5,6,7]. Paradoxically, to varying degrees, proximal tubule cells proliferating after AKI fail to dedifferentiate, undergo G2/M cell cycle arrest, and become pro-inflammatory and pro-fibrotic, which has been called failed repair of PT. Zheng et al. [8] determined whether PT cell repair successfully plays a vital role in the pathophysiology of CKD, and the failed repair of PT cells tremendously promoted AKI to CKD progression. Nevertheless, the underlying mechanisms remain unclear. 

Recently, more and more studies have shown that failed PT cell repair after AKI is the key to the development of AKI to CKD [9]. Therefore, studying the failed repair mechanisms in depth after AKI is of great clinical significance for finding effective intervention targets aiming to prevent or delay the transition from AKI to CKD. Sufficient evidence shows that oxidative stress further deteriorates PT cell failed repair and enhances AKI and CKD progression [5,10]. In AKI, oxidative stress triggers a series of cellular events, including the increased release of cytokines, infiltration of inflammatory cells, and fibroblast activation [8,11,12]. Previous studies found a large amount of macrophage infiltration around damaged PT cells after AKI, which was highly related to prognosis. However, due to the limitations of the techniques in identifying cell subsets in tissues, the potential interaction between infiltrated macrophages and impaired PT cells was unclear. Until recent years, studies gradually appeared with the development of Single-Cell RNA Sequencing (snRNA-seq). Kirita et al. [13] indicated that leukocytes played a key role in AKI–CKD progression through the Ccl2–Ccr2 ligand–receptor, and Tang et al. [14] reported that ligand–receptor signaling pathways between macrophages and PT cells in IgA nephropathy were abnormally active, indicating that macrophage infiltration might be a deteriorating risk factor for damaged PT cells. However, the underlying mechanisms through which macrophages promote AKI deterioration and CKD progression remain unclear. A better understanding of the molecular mechanisms underlying intercellular communication between macrophages and PT cells during AKI–CKD progression is critical for developing more useful therapies.

In the current study, we performed snRNA-seq on multiple mouse kidney tissues. Mice were sacrificed in sham, 3 d, and 14 d groups after 45 min of unilateral ischemia–reperfusion injury (UIRI). To ensure the accuracy of our study, we used public scRNA and spatial transcriptomic databases. We first found, in this study, that some inflammatory and oxidative stress pathways, like the IL-17 signaling pathway, were highly expressed in PT cell clusters associated with failed repair. Moreover, abnormal intercellular communication between macrophages and PT cells was essential for PT cell failed repair through CD74. We emphasized the differences between the success and failure of repair in the AKI–CKD process based on multiple factors and connected these factors with cell subtypes in the kidney, which will lead to a deeper cognition of the AKI–CKD process. 

## 2. Results

### 2.1. SnRNA and ScRNA Profiling of Kidney Cells from AKI to CKD

We developed UIRI mouse models and constructed snRNA-seq profiles to further research the mechanisms promoting failed repair of the PT (Appendix A). The flow of these processes is shown in Figure 1A. Twenty-two thousand and sixty-eight cells were isolated from three samples (8694 from the sham sample, 6038 from the 3rd day group, and 7336 from the 14th day group) after cell quality control. nCount_RNA and nFeature_RNA represented the number of unique molecular identifiers and genes, respectively, while percent.mt meant the proportion of mitochondrial genes in all genes (Figure 1B–D). The batch effect taken from different samples was reduced using “Harmony” (Figure 1E,F). We applied UMAP to visualize the cell clusters in cell projection clustering (Figure 1G). Moreover, fourteen cell types were identified according to the typical marker genes provided by Kirita et al. [13] (Figure 1H). 

### 2.2. PT Cell Subsets from AKI to CKD Progression

To research the characteristic changes in PT cell subsets, we divided PT cells into 7 PT subtypes based on specific subtype markers [13] (Figure 2B), including healthy S1, healthy S2, healthy S3, injured S1/S2, injured S3, successful repair PT, and failed repair PT (Figure 2A). We performed GSVA for these PT cell subsets. The results revealed that the IL-17 signaling pathway and D-glutamine and D-glutamate-metabolism were more active in failed repair PT cell subtypes than in other subgroups (Figure 2C). For further analysis, the subgroups with similar properties were merged to form larger subgroups to make the results easier to interpret (Figure 2D). In detail, healthy S1, healthy S2, and healthy S3 cells were merged as healthy PT, while injured S1/S2 and S3 were merged as injured PT groups. The final relative proportion of the new cell types we defined in each sample was separately calculated (Figure 2E). We noticed that the failed repair PT subset mainly appeared in 14 d after injury. 

For a further investigation of the differentiation trajectories of PT cells after injury, we performed pseudotime analysis of injured PT, failed repair, and successful repair PT cell subtypes by creating a development trajectory based on “monocle”, which mapped the cell fate after injury. In Figure 3A, we found that both failed repair and successful repair PT cells Originated from injured PT cells. However, it was noted that failed repair PT formed an alternate branch in the successful repair trajectory (Figure 3B), which indicated that the cells on different branches represented various states and the cells located on the ends of different branches had greatly distinct fates (Figure 3C). To explore the characteristic changes in the evolution of PT cells, we applied ssGSEA for the cells on the development trajectory. The results revealed that PT cells of the failed repair group were more active in the oxidative stress pathways (Figure 3D–F). 

What is more, PT cells of the failed repair group exhibited more aging properties. In addition, we performed BEAM to screen the DEGs that facilitated the transition of cells toward different fates. The DEGs were classified into three clusters with totally different gene-expression characteristics (Appendix A). In cluster 2, several DEGs were significantly enriched in the oxidative stress-related pathways, including TNF, IL-17, NF-kappa B, and the NOD-like receptor signaling pathway, which were partly consistent with the results in the ssGSEA and GSVA (Figure 3G). We discovered noticeable differences between failed repair PT, successful repair PT, and injured PT groups in the transcription factor (TF) analysis. The failed repair PT group had regulon activity for Stat3 and Dnmt3a (Figure 3H), indicating a continuous-injury and oxidative-stress status for these cells.

### 2.3. Analysis of Differentiated State Characteristics

The trajectory analysis of PT cells after injury indicates that failed repair PT cells were significantly enriched in failed repair PT cells, which reveals that failed repair PT cells may be more naïve. The differentiation phenotype is an essential feature during tubule cell repair, we compared the differentiation signature of the successful repair and failed repair PT cells based on neonatal kidney single-cell RNA-seq data from GSE94333. We identified the cell types based on the unique markers from Ide et al. [15] (Figure 4A,B), while some typical differentiation markers of the PT cell subtypes were identified by Adam et al. [16] (Figure 4C–I). Across the samples, the gene signature of the top 200 genes from the failed repair PT cluster scored highly in the mature PT part (Figure 4J), indicating that failed repair PT cells seemed to be more “mature” than successful repair PT cells. To verify this conjecture, we scored the gene signature from mature PT cells in the neonatal kidney. We found that this gene signature scored more highly in healthy and failed repair PT groups (Figure 4K), which revealed that failed repair PT cells had already lost differentiation capacity. 

### 2.4. Cross-Talk between PT Cell Subsets and Immune Microenvironment from AKI to CKD

The results above indicated that failed repair PT cells showed a remarkably pro-inflammatory and oxidative stress transcriptional signature during AKI–CKD progression. To determine how the immune microenvironment affects intercellular communication within the kidney during AKI–CKD progression, we performed a cross-talk analysis between macrophages and PT cells in UIRI mice across 3 d and 14 d (Figure 5A,B). In the cellchat differential pathways analysis, it was noticed that the App–CD74 ligand–receptor pair showed higher activity and possibility between failed repair PT cells and the immune microenvironment than successful repair. What is more, we found that CD74 was significantly highly expressed in immune cells, injured PT cells, and failed repair PT cells during AKI–CKD progression (Appendix A). The heatmaps showed that the App signaling pathway applied significant influence on injured and failed repair PT cells (Figure 5C,D) across all-time points of AKI–CKD progression. The App displayed highly distinct cell-communication networks for the failed repair PT cells and successful repair PT cells and might play an essential part in the development and progression of AKI–CKD (Figure 5E,F).

### 2.5. Failed Repair PT Cells Emerge with the Oxidative Stress Status after AKI

To verify the finding that we obtained from our own snRNA, we used data from GSE161758, which contains repaired and atrophied kidney tissue samples from wild-type mice subjected to UIRI. There are a total of 3891 cells included in this research (Appendix A). We identified the PT cell clusters based on recognized markers in the study of Puthumana et al. [17] (Appendix A–E). The PT cell count was 49% in all cells (1907 cells) (Appendix A), and the number of cells in the two groups showed significant disparity (365 atrophy PT cells vs. 1542 repair PT cells) (Appendix A).

The DEGs between repaired and atrophied PT cells from GSE161758 (Appendix A) were enriched in multiple oxidative stress-related pathways. These pathways were significantly upregulated in atrophy PT cells compared with repaired PT cells (Appendix A). The results revealed that oxidative stress was extremely active in developing kidney failed repair.

### 2.6. The IL-17 Pathway and CD74 Play a Key Role in the Progression of FR-PT Cells after Clinical AKI 

The results above indicated that abnormal oxidative stress was associated with PT cell failed repair. Therefore, we investigated whether molecularly similar failed repair PT cells can be found in human kidney samples after AKI. The human AKI cRNA-seq files from GSE145927 were selected in the next steps (Figure 6A). The cell annotation was finished according to the markers provided by Malone et al. [18] (Figure 6B), and we isolated the PT cells for the next research. We found that the IL-17 pathway-related gene signature was much More enriched in failed repair PT cells (Figure 6C,D). Moreover, CD74 was found to be expressed at higher levels in the failed repair PT cell subtype (Appendix A). These observations indicated that failed repair PT cells combined with abnormal oxidative stress is a mechanism of AKI–CKD after AKI and that CD74 might be a significant receptor during intercellular cross-talk in both mice and humans.

### 2.7. Spatial Transcriptomics Reveals the Changes after Kidney Injury

To identify where the failed repair usually occurred, we selected a series of spatial transcriptomic files from GSE182939 [19] (Figure 7A,E,I,M,Q). Figure 7 indicates that failed repair PT cells often appeared in the area where CD74 was highly expressed and CD68^+^ macrophages mainly infiltrated (Figure 7B,F,J,N,R). In addition, we found that CD74 was highly expressed in PT cells around which CD68^+^ macrophages infiltrated (Figure 7C,D,G,H,K,L,O,P,S,T) suggesting that CD74 in PT cells was not only a receptor for intercellular communication but a regulated target of macrophages. These results indicated that intercellular communication between macrophages and failed repair PT cells was at least partially dependent on CD74 expression, which might promote CKD progression after AKI.

### 2.8. CD74 Upregulation in Tubular Epithelial Cells after Co-Culture with Macrophages

We screened CD74 in Nephroseq (https://www.nephroseq.org, accessed on 10 March 2023) [20], and the database revealed that CD74 was significantly upregulated in patients with various kidney diseases (Figure 8A). In the mouse UIRI kidney models, we verified that CD74 was significantly upregulated in AKI and CKD (Figure 8B). The IHC results indicated that CD74 was expressed at a higher level in the injured and failed-repair PT cells in UIRI (Appendix A). Our fundamental experiments obtained a similar result (Figure 8C,D). What is more, we found that when co-cultured with macrophages in a hypoxic environment (Figure 8E), TECs presented with the significant upregulation of CD74 (Figure 8F,G). According to these results, macrophages might be essential for the CD74-mediated increased expression of hypoxia-treated TECs, leading to the further deterioration of PT cell failed repair, which suggested that the continuous abnormal cell communication between macrophages and TECs after injury played a more noteworthy role in promoting failed repair of PT cells and the progression of AKI–CKD.

In addition, we tested the APP levels in the macrophage cultural supernatant. As shown in Figure 8H,I, we demonstrated that hypoxia could significantly increase the mRNA expression levels of APP and stimulate macrophages to secrete APP, which might play a role in transmitting signals from macrophages to TECs during AKI to CKD progression.

## 3. Discussion

Not only AKI, a common clinical syndrome with high morbidity and mortality, but also the occurrence of CKD after AKI is accumulatively becoming a severe health issue [21]. However, the exact mechanisms of the transition from AKI to CKD remain unclear. More and more studies have shown that PT cells are the first renal intrinsic cells in the kidney microenvironment to sense damage [22,23]. The abnormal repair of PT cells and infiltration by macrophages after AKI drive the changes in renal function and structure, which is the key to promoting the progression from AKI to CKD [24]. This study aimed to reveal the failed repair of PT cell states and molecular mechanisms underlying the interaction between macrophages and failed repair PT cells.

The renal tubular epithelium contains the most sensitive cells to injury. Damage of renal tubular epithelial cells is an important cause of renal fibrosis. It has recently been recognized that AKI in humans accelerates CKD progression, with tubulointerstitial fibrosis resulting from the failed repair of PT cells after AKI. Our findings implied that hypoxia and oxidative stress might be explanations for PT cell maladaptive repair. In the clinic, we recommend that nephrologists should apply renal biopsy bulk-sequencing for severe AKI patients and evaluate the counts of failed repair PT cells with the methods and markers mentioned in our research. Nephrologists are able to pay more attention to the group of patients at a high risk of CKD progression, which fits with the current strategy for precision medicine.

Based on different kinds of scRNA-seq data from public datasets and our snRNA-seq samples, we determined novel mechanisms regulating PT cell states after AKI that underlie PT cell failed repair during AKI–CKD progression. The detailed characterization of failed repair PT cells based on the snRNA-seq cell map revealed that abnormal oxidative stress during AKI–CKD progression played a crucial role in PT cell failed repair. In addition, failed repair PT cells significantly upregulated some receptors based on the transcriptomics data, including CD74 for unusual cell communications between failed repair PT cells and macrophages. The complementary effect of the factors above promoted PT cell failed repair, with cells failing to dedifferentiate and recover normal function. Our research collectively advanced the understanding of the roles inflammation and oxidative stress play during AKI–CKD progression. One of the reasons for the progression of irreversible damage after removing the initial injury might be the abnormal inflammatory response and cell communication in the kidney. 

This study reported that some oxidative stress genes led to the dramatic deterioration of the failed repair progression of PT cells after AKI. Several of them, like Birc3, Birc2, and Ror1, were first reported to trigger the AKI–CKD progression. In addition, we found that these genes were much enriched in the TNF signaling pathway, IL-17 signaling, NF-kappa B signaling pathway, and NOD-like receptor signaling pathway. IL-17, an oxidative stress cytokine involved in innate and adaptive immune responses, is closely associated with various diseases [25]. The IL-17 signaling pathway is also able to activate the NF-kappa B pathway by increasing chemokine expression and is highly active in renal transplant recipient PT cells with acute antibody-mediated rejection [26,27]. TNF, a member of the tumor necrosis factor superfamily, possesses solid pro-inflammatory properties that are involved in apoptosis and cell differentiation [26]. In our research, the TNF signaling pathway, IL-17 signaling, and NF-kappa B signaling pathway were continuously activated in the PT associated with failed repair. Here, we declared that the activation of IL-17 might be a potential pathogenic factor and prognostic indicator in kidney ischemia/reperfusion injury.

Interestingly, we found that some chemokine genes, including Cxcl1 and Cxcl2, were highly expressed in the failed repair PT. What is more, the NOD-like receptor signaling pathway was also enriched in the failed repair PT cells. Previous studies had proved that some hub genes in this signaling pathway were associated with the clinical prognosis of CKD patients [28,29]. Therefore, our findings might provide novel insight into the relationship between the NOD-like receptor signaling pathway and CKD. 

We further screened the potential regulatory TFs that regulate PT cell failed repair after AKI. The SCENIC results suggested that Stat3 and Dnmt3a might be hub TFs involved in the regulation of the failed repair PT. According to previous studies, Stat3 is involved in the active infiltration of immune cells in tumors and Alzheimer’s disease [30]. Fletcher et al. [31] reported that the inhibition of Stat3 would reduce the number of infiltrating macrophage in the tumor. It was declared that Stat3 promoted pulmonary fibrosis by mediating IL17 and TGF-beta production [32]. Our research emphasized the role of Stat3 in oxidative stress and fibrosis during AKI–CKD progression. Dnmt3a was associated with increased cardiovascular events and oxidative stress enhancement [33]. Li et al. [34] declared that Dnmt3a expression was closely associated with macrophage polarization. Here, we showed that Dnmt3a might be a hub TF regulating PT cell differentiation after AKI.

The differences in the cell communication between PT cells with different fates and macrophages were observed in our snRNA-seq analysis, revealing that this cell communication might play an essential role in CKD progression. We evaluated the intercellular signaling network by analyzing the receptor–ligand interactions during AKI–CKD progression. The App–CD74 signaling axis was observed in cell communication between failed repair PT cells and macrophages across all time points from AKI to CKD. Firstly, amyloid precursor protein (APP) is of great interest to the researchers focusing on Alzheimer’s disease [35]. After years of research, APP was found to play a role in the central nervous system and peripheral tissues of the liver and pancreas, adipose tissue, and myotubes [36]. CD74 was evidently associated with T cell development and macrophage inflammation. Valiño-Rivas et al. [37] declared that CD74, a macrophage migration inhibitory factor (MIF) receptor, was highly expressed in PT cells in various nephropathies. MIF increased inflammatory cytokines through the activation of CD74. In line with the study by He et al. [38], our study found the potential that the APP–CD74 signaling pathway axis is extensively activated in AKI based on the transcriptome. Our findings might provide novel insights into the function of CD74 during AKI–CKD progression.

TECs significantly upregulated CD74 in kidney disease patients and kidney-injured mice in vivo [20,37]. However, the bulk-seq DEG results and the fundamental experiment that we performed proved that CD74 was not significantly upregulated in the pure hypoxic TECs. We found that TECs co-cultured with macrophages in a hypoxic environment would significantly increase CD74. These results revealed that macrophages were necessary to regulate the expression of CD74 in TECs. In addition, the intercellular communication between macrophages and TECs through APP–CD74 markedly aggravated kidney damage and promoted failed kidney repair. Thee abnormal pathological processes form a loop that might be a therapeutic target to prevent failed kidney repair and CKD after AKI.

## 4. Materials and Methods

### 4.1. Animals 

The Animal Experiment Administration Committee of the Fourth Military Medical University approved all animal experiments. In this study, we selected C57BL/6 male mice weighing 22–25 g at 8–10 weeks. After being anesthetized with isoflurane, mice were subjected to renal UIRI using a proven approach, as previously described [39]. We performed ischemia using a microvascular clamp-on only on one side of the renal pedicle for 45 min at 37 °C. Then, we removed the microvascular clamps, and the kidneys were returned to the peritoneal cavity after finishing UIRI. Sham operations only included blunt separation of the kidney without any induction of ischemia. The incisions were closed with a suture. All surgical procedures were performed by Author Ying Zhou, a specialized staff member in experimental surgery. All animals involved in the experiments were euthanized via excess carbon dioxide. 

### 4.2. Mouse Kidney Samples

The mice were euthanatized and transcardially perfused with cold PBS. In this research, we selected the UIRI model as our research model. UIRI is a classic and recognized AKI–CKD model, which represents the ischemic etiology of AKI. Then the kidney samples were harvested from the sham group and 3 d and 14 d after UIRI. All kidney samples for snRNA-seq were snap-frozen with liquid nitrogen.

### 4.3. 10× Genomics-Based snRNA-seq

The 10× Genomics platform uses microfluidic technology to wrap beads and cells with a Cell Barcode in droplets, collect the droplets with cells, and then lyse the cells in the droplets so that the mRNA in the cells and the cells above the bead are lysed. The barcode is combined to form Single-Cell GEMs, a reverse transcription reaction is performed on the droplets, a cDNA library is constructed, and the sample source of the target sequence is distinguished by the sample index based on the library sequence. 

We applied trypan blue staining (Sigma-Aldrich, Darmstadt, Germany) to ensure that cell viability was more than 90%. Freshly stored kidney cells were processed for library preparation and sequencing by OE Biotech (Shanghai, China) Co., Ltd. according to 10× Genomics guidelines. De-multiplexing, read alignment and quality control were performed at OE Biotech (Shanghai) Co., Ltd. using the 10× Cell Ranger package (v6.1.2), as well. 

### 4.4. snRNA-seq Data Processing 

The Seurat R package (version 4.0.5) was applied for the downstream data analysis [40]. The “Seurat” package makes use of the Wilcoxon rank sum test. In our snRNA data, we filtered out cells with fewer than 200 genes or more than 7000 genes. In addition, the cells with more than 5% mitochondrial genes were filtered out. The “LogNormalize” method was carried out for gene expression normalization. Two thousand highly variable genes (HVGs) were selected using the “vst” method based on data normalization. Principal components were identified through principal components analysis. We applied the “harmony” R package 0.1.0 to avoid the batch effect taken from different samples. We set the resolution as 0.6 for the “FindClusters” function to classify the kidney cells into sub-clusters. For screening marker genes of each cluster, an adjusted *p*-value < 0.05 and |log fold-change| > 0.25 were set as a cut-off for the “FindAllMarkers” function. In this study, we manually identified the cell type for every cluster based on the marker genes we obtained from previous studies [13,16,17,18,19].

The public scRNA files for human and mouse kidney tissues were from the Gene Expression Omnibus (GEO). Repaired and atrophied kidney tissues were from GSE161758. The scRNA files for neonatal kidneys were obtained from GSE94333. Human injured kidney sample data were from two clinical tubular injury biopsy samples without any evidence of immunologic injury. The analysis flow was similar to the steps mentioned above. The only differences were that we needed to follow the criteria for the cut-off and markers in their related citations [13,16,17,18,19]. We added the marker list to Appendix A.

### 4.5. Pseudotemporal Analysis

We used the “Monocle” R package (v2.18.0) for the pseudotemporal analysis [41]. PT cells were ordered onto a trajectory based on HVGs identified in the time-course analysis. In the next step, we screened the branch-dependent genes in each branch using branched expression analysis modeling (BEAM). Then, these genes were classified into 3 clusters following a pseudotime manner. We performed a KEGG analysis of the hub gene clusters for further research.

### 4.6. Cell Communication Analysis

To study the cross-talk between cells in the kidney, we applied the “CellChat” R package (v1.6.0), widely used to analyze the intercellular communication networks in the snRNA-seq data [42]. “CellChat” uses network analysis and pattern recognition methods to predict the key signal inputs and outputs of cells and how those cells and signals coordinate functions. By default, the “CellChat” package uses a statistical method called “trimean”. “CellChat” classifies signaling pathways across different datasets and characterizes conserved and context-specific pathways through multiple learning and quantitative comparisons.

### 4.7. Gene Enrichment Analysis

Kyoto Encyclopedia of Genes and Genomes (KEGG) and Reactome pathway enrichment analyses were performed using webgestalt (http://www.webgestalt.org/, accessed on 10 February 2023) [43] and KOBAS (http://bioinfo.org/kobas, accessed on 11 February 2023) [44]. For the gene enrichment analysis, a hypergeometric test was used. We set a *p*-value < 0.001 as a cut-off for the significantly enriched pathways. The “GSVA” function in R was used to perform ssGSEA to evaluate the enrichment of each KEGG pathway. 

### 4.8. Cell Type Correlation Analysis

We performed correlation analysis between two cell types in two independent databases based on a series of marker genes for each cell cluster. The marker genes were screened using the “FindAllMarkers” function in R, and the criteria were an adjusted *p*-value < 0.05 and |log fold-change| > 0.25. The top 200 marker genes were used to construct a labeled gene signature for each cell type, which could be applied to calculate a cell type module score. Then, we calculated the Person correlation between the module score of targeted cell types and the module score of all the cell types in another scRNA-seq dataset. We could explore the relationship between different samples and databases through this method [45]. For more details about the module score calculation, please see the Appendix A.

### 4.9. Pathway Annotation Analysis

The gene expression profiles of targeted pathway gene sets are the basis of pathway annotation analysis [46]. We calculated each gene in the targeted pathways for a score based on gene expression in each barcode with the “GSVA” function in the R package. The score represents the activity of the targeted pathway, which could be used to evaluate pathway enrichment levels between different cell clusters [47]. 

### 4.10. Spatial Transcriptomics

The spatial objects were loaded into Seurat in R. Data normalization was performed through the “SCTransform” function, and we matched the gene expression and samples to ensure that numerous genes and numerous samples could be visualized simultaneously. For the concision of our analysis, recognized cell markers were used to identify the distribution of corresponding types of cells.

### 4.11. Cell Culture

BUMPT (a mouse renal tubular epithelial cell line) and RAW264.7 (a mouse mononuclear macrophage leukemia cell line) cells were cultured in Dulbecco’s modified Eagle medium (DMEM) with 10% fetal bovine serum (FBS) and a 1% penicillin-streptomycin (P-S) mixture. All culture media and FBS were obtained from Biological Industries Co. Ltd. The HK2 (human renal cortical proximal tubule epithelial) cell line was obtained from the American Type Culture Collection (ATCC number: CRL-2190). Ordinary cultured cells were exposed to a humidified atmosphere consisting of 5% CO_2_ and 95% air (20% O_2_). Hypoxia-treated cells were exposed to 5% CO_2_, 1% oxygen, and 95% nitrogen.

### 4.12. Co-Culture

We constructed a co-culture system with 0.4 μm polycarbonate membrane Transwell Permeable Supports (Corning, Tewksbury, MA, USA). The co-culture system separated RAW264.7 and BUMPT cells into different compartments. Two hours prior to co-culture, 10^4^ BUMPT cells were seeded on a 24-well plate, and then, 10^4^ RAW267.4 cells were placed into the top chamber of a transwell insert. The co-culture lasted 48 h. We used DMEM with 10% FBS and a 1% P-S mixture for the co-culture without other supplements.

### 4.13. Quantitative Real-Time PCR (qRT-PCR)

RNA from cells was isolated using RNAiso Reagent (Takara Bio Inc, Otsu Prince, Japan). qRT-PCR was performed using SYBR qPCR Master Mix (Vazyme, Nanjing, China) and a Roche LC480 Real-Time PCR System (Roche, Basel, Switzerland). We used the 2^−ΔΔCt^ method to evaluate the RT-PCR [48].The primer sequences used in this research are listed in Appendix A. 

### 4.14. Western Blotting

Mouse renal tissues and cells were lysed with RIPA lysis buffer containing protease inhibitors, according to a standard procedure [49]. We measured the protein concentration using a Bicinchoninic Acid Assay (BCA). Thirty micrograms of protein from each sample was loaded every time. After being blocked, the membranes were incubated with antibodies against CD74 (1:1000, Servicebio, Wuhan, China) and β-actin (1:5000, CST, Boston, MA, USA) for 12 h at 4 °C. Eventually, the members were incubated with secondary antibodies at room temperature for 1 h. All experiments were repeated three times. 

### 4.15. Enzyme-Linked Immunosorbent Assay (ELISA)

The macrophage culture supernatants were centrifuged (12,000× *g*; 5 min; 4 °C), and the supernatant was fractionated and stored in a −80 °C freezer. Amyloid precursor protein (APP) was measured using a mouse ELISA kit (Shanghai Jianglai Biotech, Shanghai, China). All the steps strictly followed the manufacturer’s instructions. 

### 4.16. Immunohistochemistry (IHC)

We performed immunohistochemical staining on 4 µm-thick tissue slides. After deparaffinization, we applied 1 mM ethylene diamine tetraacetic acid (EDTA), pH 9.0, for antigen retrieval in a steam cooker. Then, we applied CD74 primary antibodies (1:150, Servicebio, Wuhan, China) at 1:100 in antibody diluent for 1 h. Then, the slides were incubated with HRP-conjugated secondary antibody for 15 min. We applied diaminobenzidine before mounting, and the slides were counterstained with hematoxylin.

## 5. Conclusions

In conclusion, our research applied a variety of snRNA-seq and scRNA-seq data from mouse models and patients combined with multiple methods, such as pseudotemporal analysis, cell communication analysis, ssGSEA, and so on, to determine the underlying mechanism promoting AKI–CKD progression. Our studies showed that the regulation of failed repair PT cells and the interaction between PT cells and the immune microenvironment seemed to play critical roles during AKI–CKD progression. We presented a series of TFs and oxidative stress-related genes that might promote CKD progression after AKI. CD74 was identified as a potential marker of the failed repair PT cells. We intended to reveal that the new potential mechanism of the transition from AKI to CKD was caused by the interaction with the immune microenvironment and provided new ideas and new targets for the prevention and treatment of CKD.

### Limitations

Limitations in our study include, besides the lack of biological duplicates in the snRNA data analysis, the exact mechanisms through which the App–CD74 axis causes injured PT cell failed repair are still not entirely clear. Notwithstanding, our research revealed a series of quite interesting conclusions and our findings deserve more work in the future.

## Figures and Tables

**Figure 1 ijms-24-11617-f001:**
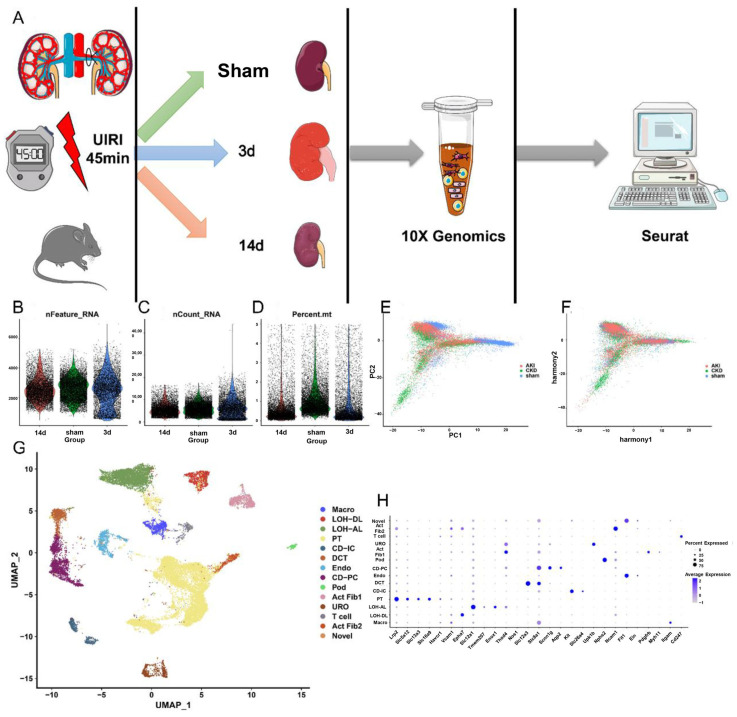
Single-nucleus RNA sequencing (snRNA-seq) identifies dynamic cellular state transitions of tubular epithelial cells after severe IRI. (**A**) Drop-seq strategy. UIRI, unilateral IRI (**B**) The genes (features), (**C**) counts, and (**D**) mitochondrial gene percentage of each sample. Each plot represents a cell. The PCA distribution of each cell in each kidney sample (**E**) before normalization and (**F**) after normalization. Different colors represent different cell sources. (**G**) Fourteen distinct cell clusters were identified based on UMAP plotting, and clusters were colored and labeled distinctively. The color of cells represents the group origin. (**H**) Dot plot of cell-type marker genes. The color of dots represents average expression, and the size of dots represents the average percent of cells expressing the selected genes. Abbreviations are as follows: CD-PC, principle cells of collecting duct; DCT, distal convoluted tubule; Pod, podocytes; LOH-DL, descending limb of loop of Henle; Endo, endothelial cells; Act Fib 1, active fibroblasts 1; Act Fib 2, active fibroblasts 2; CD-IC, intercalated cells of collecting duct; Macro, macrophages; LOH-AL, thin ascending limb of loop of Henle; URO, urothelium. PT, proximal tubule.

**Figure 2 ijms-24-11617-f002:**
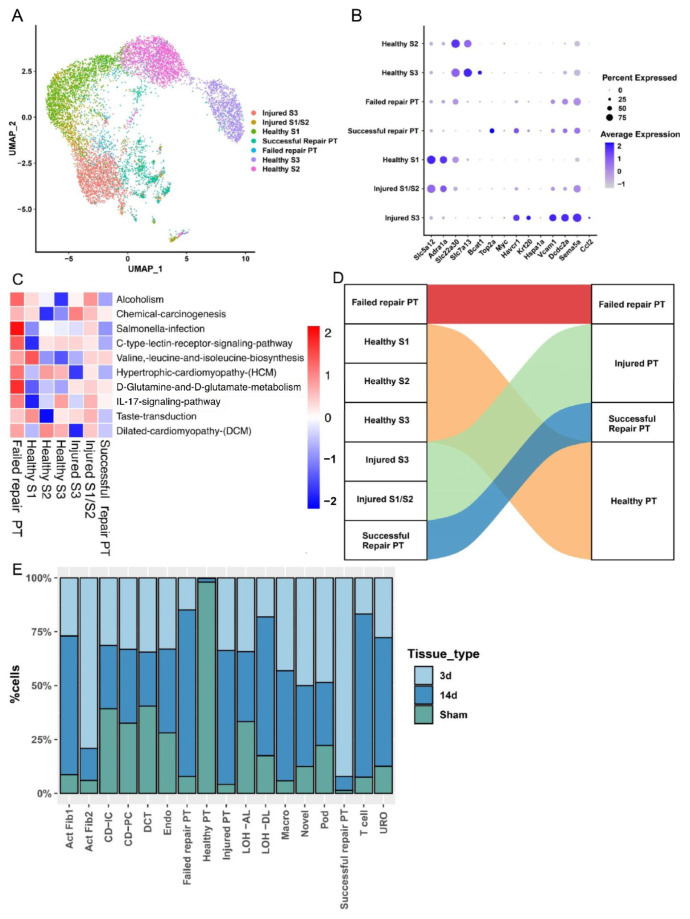
Failed repair PT cells show an inflammatory and oxidative stress transcriptional signature and mainly appear in the period of chronic injury. (**A**) UMAP of 7 different PT clusters, including injured S3, injured S1/S2, healthy S1, healthy S2, healthy S3, successful repair PT, and failed repair PT. (**B**) Dot plot of cell type marker genes. The color of the dots represents the average expression, and the size of the dots represents the average percentage of cells expressing the selected gene. (**C**) The GSEA (Gene Set Enrichment Analysis) heatmap result shows the top10 differential enriched KEGG (Kyoto Encyclopedia of Genes and Genomes) pathways in each PT subtype. All pathways shown in this figure occupied significant differences compared with other subtypes (*p* < 0.05). (**D**) A Sankey diagram shows the results of the PT re-grouping. For brevity, the cells with similar features are re-clustered. Healthy S1, healthy S2, and healthy S3 are grouped into a new group named healthy PT. Injured S3 and injured S1/S2 are grouped into a new group called injured PT. After re-grouping, the bar chart (**E**) shows each kidney sample count in every cell cluster. The different colors represent different kidney samples.

**Figure 3 ijms-24-11617-f003:**
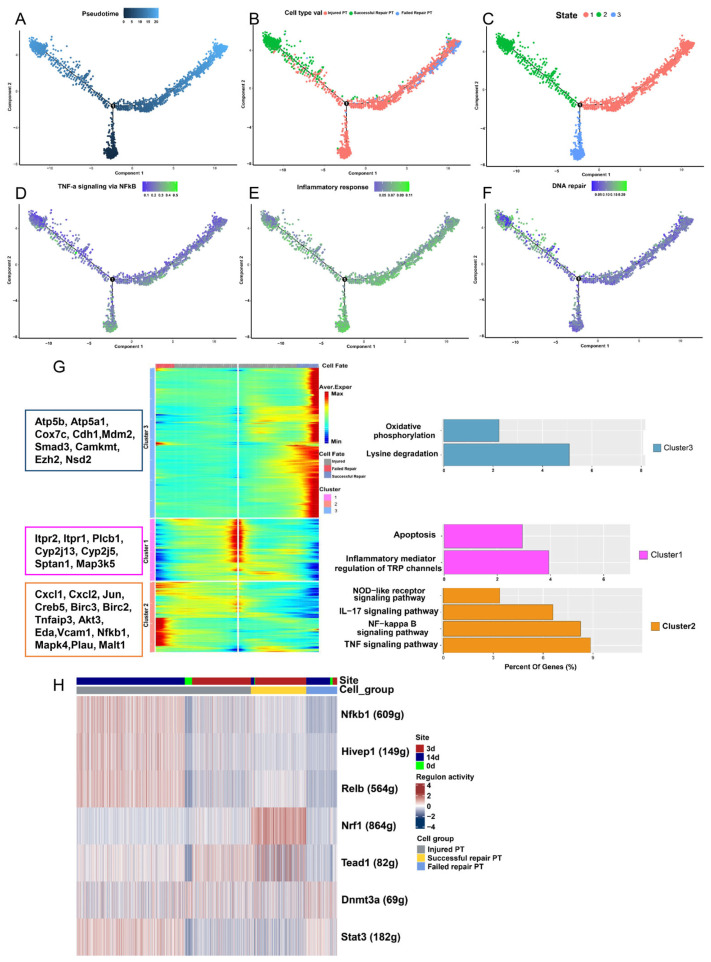
Trajectory analysis of PT cells after injury (failed repair PT, successful repair PT, and injured PT), and oxidative stress characteristics of the failed repaired PT cell subset. (**A**) Monocle2 pseudotime trajectory of proximal tubular cells, which were colored based on the pseudotime. (**B**) Monocle2 pseudotime trajectory of proximal tubular cells, which were colored based on cluster identity. (**C**) Monocle2 pseudotime trajectory of proximal tubular cells, which were colored based on the cell state. (**D**–**F**) UMAP rendering of signaling pathways. (**D**) Note that TNF-α signaling via the NF-κB pathway was enriched in both injured PT cells and failed repair PT cells. (**E**) Note that the inflammatory response pathways were enriched in injured PT cells and failed repair PT cells. (**F**) Note that the DNA repair pathway was enriched in the failed repair PT cells. (**G**) Pseudotemporal gene-expression profiles of some DEGs in PT cell subsets, excluding the ribosomal and mitochondrial genes. Right bar diagrams display the enriched KEGG terms of the three gene modules. (**H**) The heatmap shows that transcription factors (TFs) that are significantly associated with the development of different PT cell subsets. The number in the brackets behind the genes represents the target gene number regulated by the transcription factors.

**Figure 4 ijms-24-11617-f004:**
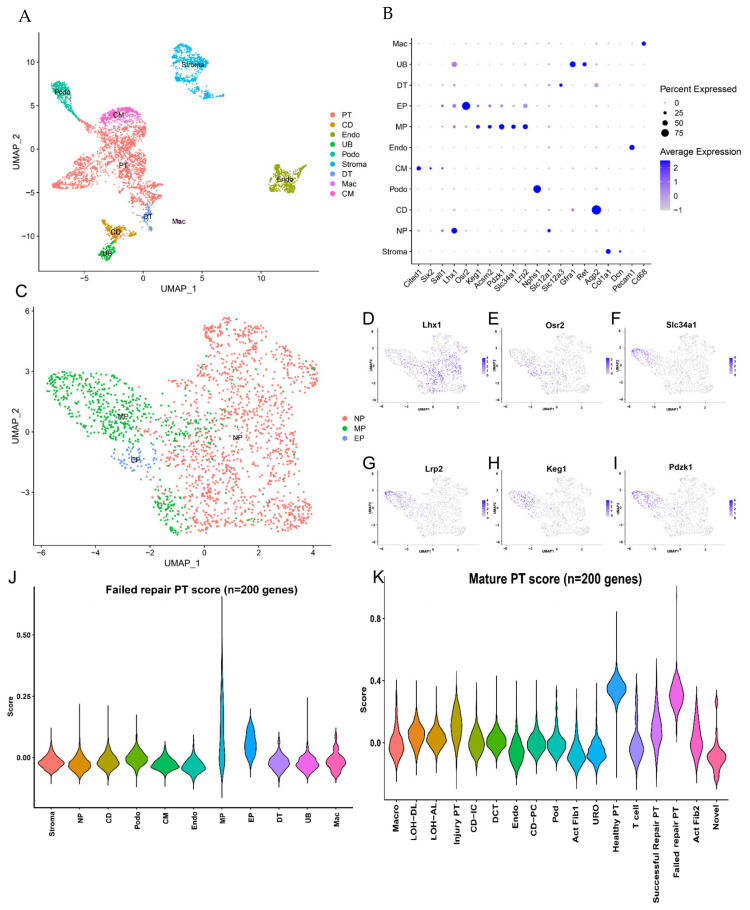
Comparative analyses of failed repair PT cells and neonatal proximal tubular cells. (**A**) UMAP plots show mouse neonatal kidney cells from post-natal day 1 kidney. (**B**) Dot plots show the gene expression patterns of cluster-enriched canonical markers. (**C**) UMAP plots show mouse PT cells from neonatal kidney cells. The PT cells were classified as NP, EP, and MP. Abbreviations are as follows: NP, nephron progenitor; EP, early PT; MP, mature PT. Expression of nephron progenitor cell marker (**D**) Lhx1, early PT cell marker (**E**) Osr1, and nephron progenitor cell markers (**F**) Slc34a, (**G**) Lrp2, (**H**) Keg1, and (**I**) Pdzk1. (**J**) Violin plots of failed repair PT scores of individual spots derived from AKI-CKD scRNA-seq data for each cluster. (**K**) Violin plots of mature PT scores of individual spots derived from neonatal kidney scRNA-seq data for each cluster. Dotted boxes outline clusters with the highest average. Abbreviations are as follows: CM, cap mesenchyme. NP, nephron progenitor. EP, early PT. MP, mature PT. Podo, podocyte. DT, distal tubule. UB, uretic bud. CD, collecting duct. Endo, endothelial cells. Mac, macrophages.

**Figure 5 ijms-24-11617-f005:**
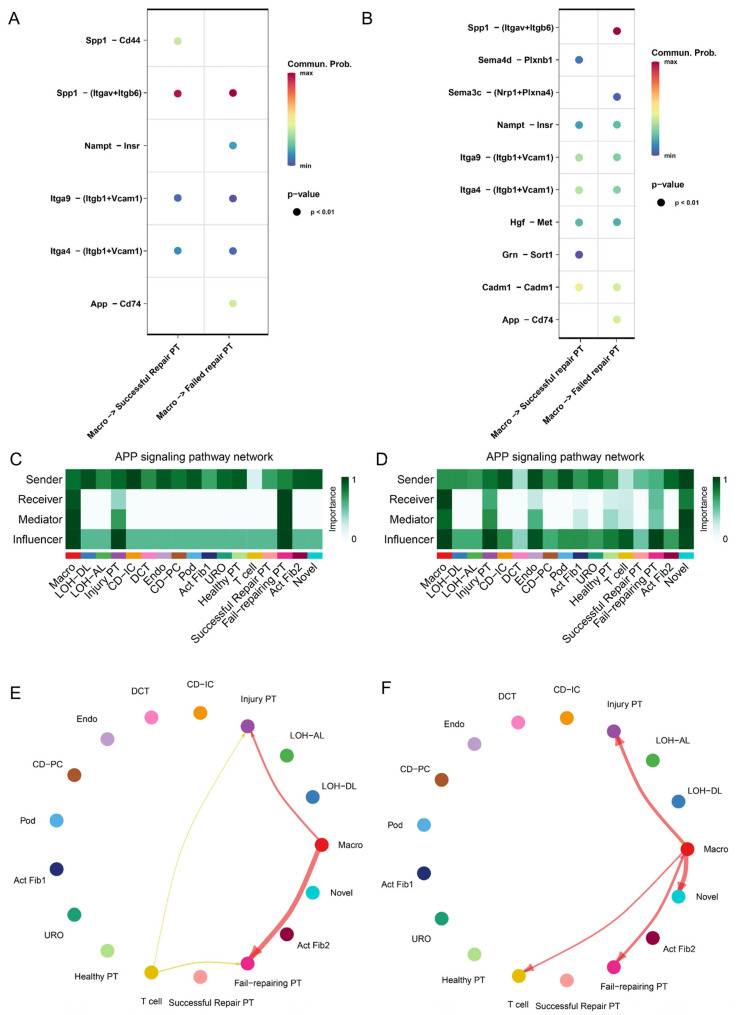
Cell communication analysis between PT cells and macrophages in different kidney samples. (**A**) Dot plot of different cell communication pathways in 3 d kidney samples. (**B**) Dot plot of different cell communication pathways in 14 d kidney samples. The color of dots represents the possibility of the existence of the pathway, and the size of dots means that there are significant differences in the pathway based on the cell communication analysis. (**C**) The heatmap shows the importance of the APP-signaling pathway network in different cell clusters in UIRI 3 d kidney samples. (**D**) The heatmap shows the importance of the App-signaling pathway network in different cell clusters in the UIRI 14 d kidney sample. (**E**) An overview of the App–CD74 signaling pathway in PT cell–immune microenvironment interactions in UIRI 3 d kidney samples. (**F**) An overview of the App–CD74 signaling pathway in PT cell–immune microenvironment interactions in UIRI 14 d kidney samples. The arrow and edge color indicate direction. Edge thickness indicates the weighted interaction between cell clusters.

**Figure 6 ijms-24-11617-f006:**
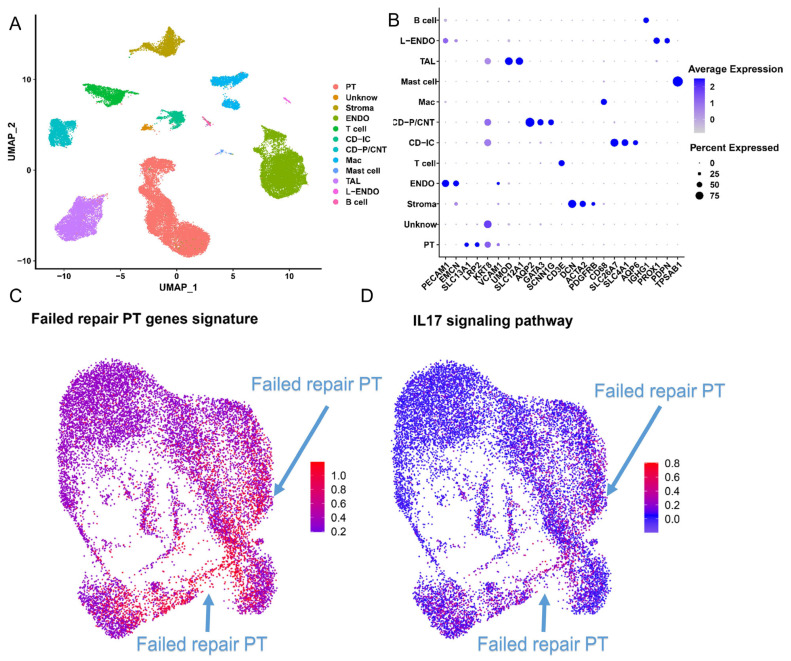
Characterization of single-cell RNA-seq data from human injured kidney samples. (**A**) UMAP plots showing human injured kidney samples without rejection. (**B**) Dot plot showing the gene expression patterns of cluster-enriched typical markers. The color of the dots represents the average expression, and the size of the dots represents the average percent of cells expressing the selected gene. (**C**) UMAP plots show the failed repair PT gene signature expression in the human injured kidney samples. Arrowheads: failed repair PT gene is highly expressed. (**D**) UMAP rendering of signaling pathways. Note that genes for IL-17 pathway signaling were highly expressed in the failed repair PT group. The plots that scored highly in the failed repair PT gene signatures are circled. The arrows show the location of failed repair PT cells in UMAP.

**Figure 7 ijms-24-11617-f007:**
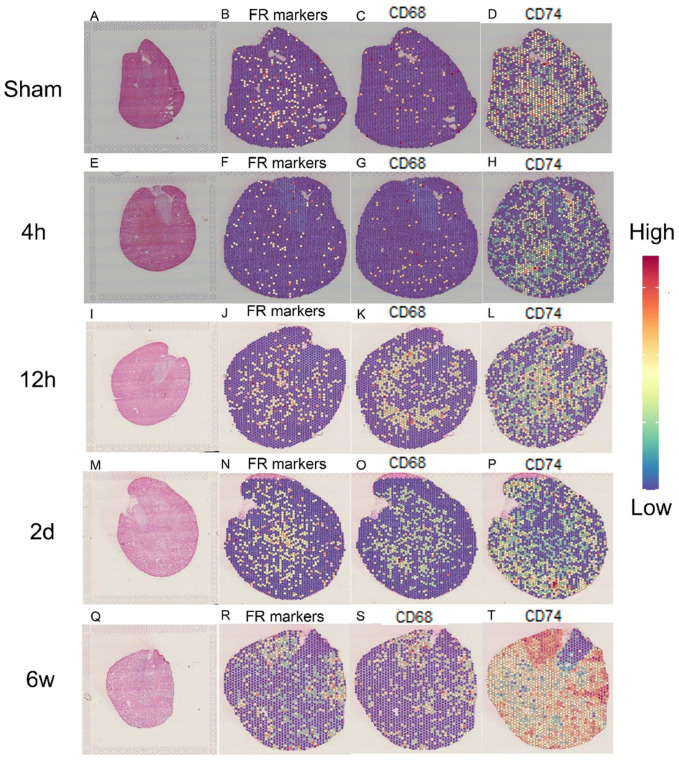
Resolving spatial relationships of cell types and gene expression with Seurat. (**A**) Tissue section of the sham kidney. (**B**) Remaining tissue-covered spots expressing failed repair PT cell markers in the sham kidney. (**C**) Remaining tissue-covered spots expressing CD68^+^ macrophage markers in the sham kidney. (**D**) Remaining tissue-covered spots expressing CD74 in the sham kidney. (**E**) The tissue section of IRI at 4 h in the kidney. (**F**) Remaining tissue-covered spots expressing failed repair PT cell markers 4 h post-injury. (**G**) Remaining tissue-covered spots expressing CD68^+^ macrophage markers 4 h post-injury. (**H**) Remaining tissue-covered spots expressing CD74 4 h post-injury. (**I**) The tissue section of IRI at 12 h in the kidney. (**J**) Remaining tissue-covered spots expressing failed repair PT cell markers 12 h post-injury. (**K**) Remaining tissue-covered spots expressing CD68^+^ macrophage markers 12 h post-injury. (**L**) Remaining tissue-covered spots expressing CD74 12 h post-injury. (**M**) The tissue section of IRI at 2 d in the kidney. (**N**) Remaining tissue-covered spots expressing failed repair PT cell markers 2 d post-injury. (**O**) Remaining tissue-covered spots expressing CD68^+^ macrophage markers 2 d post-injury. (**P**) Remaining tissue-covered spots expressing CD74 2 d post-injury. (**Q**) The tissue section of IRI at 2 d in the kidney. (**R**) Remaining tissue-covered spots expressing failed repair PT cell markers 2 d post-injury. (**S**) Remaining tissue-covered spots expressing CD68^+^ macrophage markers 2 d post-injury. (**T**) Remaining tissue-covered spots expressing CD74 2 d post-injury. FR markers, failed repair markers represent Vcam1, Dcdc2a, and Ccl2.

**Figure 8 ijms-24-11617-f008:**
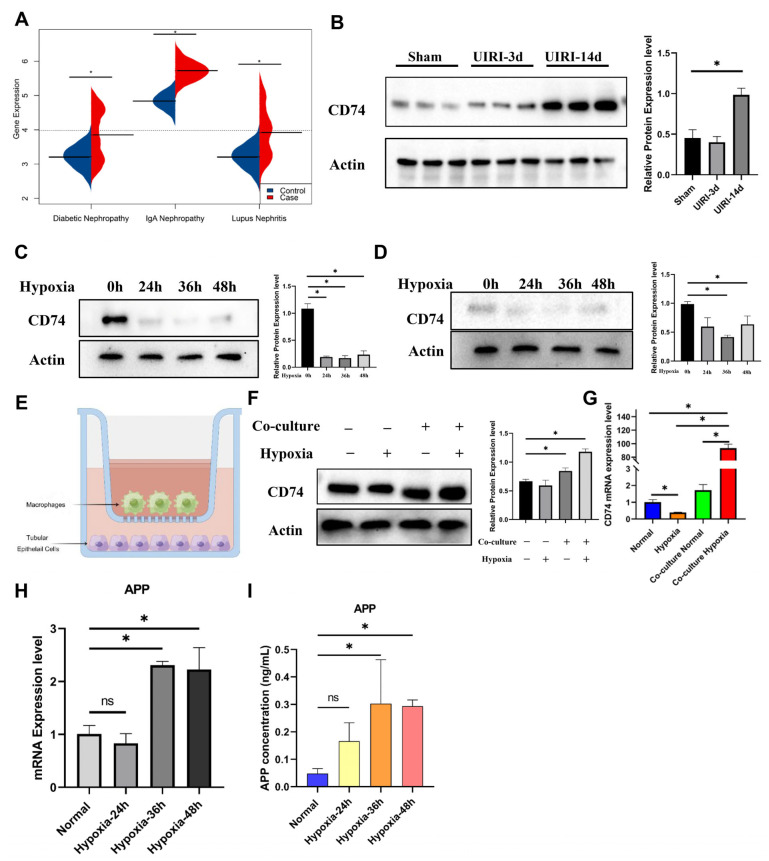
Verification of the expression of CD74 in vivo and in vitro. (**A**) CD74 was significantly upregulated in micro-dissected kidney tissue in the CKD cohort that consisted of lupus nephritis (LN), diabetic nephropathy (DN), and IgA nephropathy (IgAN). (**B**) Western blotting verified that CD74 was upregulated in mouse UIRI kidney tissues. (**C**) Western blotting verified that CD74 did not increase significantly in hypoxia-stimulated HK-2 cells. (**D**) Western blotting verified that CD74 did not increase significantly in hypoxia-stimulated BUMPT cells. (**E**) Schematic drawing of co-culture system of BUMPT cells and RAW264.7 cells. (**F**) BUMPT cell CD74 protein levels were significantly upregulated in a hypoxic co-culture system with macrophages compared with those under pure hypoxia, and (**G**) BUMPT cell CD74 mRNA levels were significantly upregulated in a hypoxic co-culture system with macrophages compared with those under pure hypoxia. (**H**) qRT-PCR found that APP was significantly upregulated in hypoxia-stimulated macrophages. (**I**) ELISA found that APP was significantly upregulated in hypoxia-stimulated macrophage supernatants. ns means no significance. * represents *p* < 0.05.

## Data Availability

The raw data supporting the conclusions of this article will be made available by the authors without undue reservation. The data in the current study are available from the corresponding author upon reasonable request. And the public data can be acquired below: https://www.ncbi.nlm.nih.gov/geo/query/acc.cgi?acc=GSE161758 (accessed on 2 January 2023), https://www.ncbi.nlm.nih.gov/geo/query/acc.cgi?acc=GSE94333 (accessed on 2 January 2023), https://www.ncbi.nlm.nih.gov/geo/query/acc.cgi?acc=GSE12792 (accessed on 2 January 2023), https://www.ncbi.nlm.nih.gov/geo/query/acc.cgi?acc=GSE182939 (accessed on 2 January 2023), and https://www.ncbi.nlm.nih.gov/geo/query/acc.cgi?acc=GSE145927 (accessed on 2 January 2023).

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
