# Peer review of "Cell Profiling of Acute Kidney Injury to Chronic Kidney Disease Reveals Novel Oxidative Stress Characteristics in the Failed Repair of Proximal Tubule Cells"

_ijms, 2023, doi:10.3390/ijms241411617_

Round 1

Reviewer 1 Report (Previous Reviewer 1)

This is an identical version of a previously submitted manuscript. The Authors did not add any novel experiments but slightly modified the introduction. The overall experimental designed is very confusing and the data are not clearly presented. The results showed do not support the authors'conclusions. cd74 is expressed everywhere in the kidney according to figure 7 so I don't know how they can conclude it is associated with FR cells. They performed a snRNAseq but did not provide access to the dataset nor they verified their findings with staining or realtime etc. 

The english needs to be extensively revised

Author Response

Reviewer 2 Report (New Reviewer)

I read this work with interest.

The issue studied is very relevant and of major pathophysiological interest. 

Nevertheless, I was confused by the term AKI. I think the more appropriate term would be ischemic acute kidney injury. The lesion model used in this study was an ischaemic model by kidney hilium clamping. There are many other etiology of AKI so the field of the physiopathology described in this work limitted to ischemic etiology of AKI. (DOI: 10.1038/s41419-023-05830-z)

The presentation of the results should be a little clearer in view of the number of techniques used. They are more clearly explained for example in the article cited in reference 13 by the authors.

The discussion section needs to be more nuanced. They are exploring the transcriptome but not the proteome or the functionality of proteins. There is a gap between transcriptome and function and I think they need to shade their conclusion.

Author Response

Reviewer 3 Report (New Reviewer)

Yu et al. report that novel oxidative stress might have an important role in the failed repair of proximal tubule cells in the AKI to CKD model. The authors use single-cell profiling and snRNA sequencing after the UIR injury model in mice. This study was a well-designed animal experiment.

1. After 6 weeks of UIR injury, the kidney might have change to atrophy. So, the kidney tubules have changes to fibrotic characteristics. How about the expression of fibrotic markers or EMT markers?

2. In figure 4, why do authors have comparative analyses of failed repair PT cells and neonatal proximal tubular cells? The authors need to explain more detailed reasons.

3. Authors show the crosstalk between PT cells and the immune microenvironment for AKI to CKD. In supplementary Figure 3, CD74 expression was increased in the UIR injury kidney. However, what kinds of cells are dominantly expressed in the UIR kidney? If possible, authors need to show double immunofluorescence stains with tubule-specific markers with CD74 or immune cell markers.

5. Authors need to check the legend for supplementary figure 3.

Round 2

Reviewer 1 Report (Previous Reviewer 1)

The paper has remained substantially the same. Therefore, my recommendation has not changed. 

Need to be substantially revised. 

This manuscript is a resubmission of an earlier submission. The following is a list of the peer review reports and author responses from that submission.

Round 1

Reviewer 1 Report

This paper from Yu Z. et al., employs snRNA-seq generated by the authors and from available datasets to analyze the oxidative stress of the so-called “failed repair” population. However, there are many inconsistencies across the manuscript, some major flaws and a general lack of novelty that decrease the overall enthusiasm for this paper.

1.        While the Authors generated a novel snRNA-seq dataset, they failed to provide accession code for the dataset and there is no reference for the Reviewers to check it. Moreover, no information are provided in terms of kidney functionality or tissue analysis for the UIRI model that the authors employed.

2.       The Authors used previously published markers to identify the different populations but there is no reference to which markers they actually used, therefore it is impossible to verify if what they are claiming it is actually true.

No validation of the scRNA-seq data is provided 

Reviewer 2 Report

The topic is very interesting and exposes how the pathophysiology of acute kidney injury and its progression towards chronic kidney damage could be.

It would be interesting to highlight how these discoveries can be monitored in vivo and be useful for the clinic.

Is it possible to identify a marker of renal damage and its progression that can be used in the clinic in relation to the results?

It is possible to highlight a pathophysiological connection between the markers: TIMP-2 (tissue inhibitor of metalloproteinase-2) and IGFBP7 (insulin-like growth factor binding protein 7), NGAL and CCL14. The pathways that have identified with your work can have interconnections with these molecules?

Reviewer 3 Report

In the present study Yu et al. perfomed snRNA seq analysis of different time points of the IR model to analyze the AKI-to-CKD transition. In this cell profiling they focus on differences between tubular cells which fail repair vs. successful repair and find APP-CD74 as an interesting pathway. They claimed to identify abnormal intercellular communications between macrophages and PT cell clusters leading to proximal tubular (PT) cells failed to repair AKI-CKD progression. Although the idea of the study is potentially interesting some methodological uncertainties e.g. about the n-number used for the snRNA seq approach or about the stated mechanism, which is not shown, need substantial attention. In detail:

Mechanism:

·         It is mentioned that a molecular mechanism has been shown (among others in the abstract), but at the same time it is mentioned in the limitations that no exact mechanism can be shown.

·         The regulation shown in Figure 9 should be deleted completely, as this mechanism was not shown and thus was completely speculatively merged from the previous data.

·         The data about CD74 expression seem not be in line. In the public data about CD74 in hypoxia is no different expression visible, but in the shown blots and on mRNA level, Cd74 seem to be down-regulated, could the authors comment on that

·         Macrophages are also a source of MIF, one of the main proteins binding to CD74. The present data gave only hints to an APP-CD74 interaction. To proof the other binding partners should be inhibited. Further the pathway after binding of APP to CD74 is not clear. How can this binding induce CD74 expression? And more interesting, how does this influence the TEC repair?

General issues:

·         Representations are sometimes so small that no conclusion can be drawn from the illustration, as it is simply not recognizable.

·         Abbreviations are not explained throughout the paper.

·         Figure 2. Labels of A and B are missing. Further Figure 2B is not mentioned in the text.

·         Figure 3 is completely missing from the paper but is described in the Results section.

·         In the text “Supplementary tables” are written. Are these identical with supplementary “files”, please add table heading and legend.

·         In addition, biological duplicates are missing.

·         Sometimes far insufficient data points have been used, e.g., in 8C.

·         A consistent structure should always be used in the figures such as 1st controls, 2nd sham, 3rd day 3 4th day 14. There should be no deviation from this in any figures.

Methodological issues:

Animal experiment:

·         How many mice were used? Were they divided into randomized groups?

·         Exactly what time points of the IRI were taken? Since this varies between the experiments. Once it was mentioned after 6 weeks (Figure 7)?

·         It is unclear what the group “d0” is, are these the later called sham mice? If these are sham animals, after which time point after the OP the mice have been sacrificed?

RT-PCR

  • Which method was used to evaluate the RT PCR?
  • Figure 8H: Please also insert the expression level of the Normal Co-Culture.
  • Figure 8I: Why were different time points used compared to those used for the Western blots. Please be consistent. This is also true for the ELISA.

In vitro studies:

  • Please insert which media was used for the co-culture and if supplements were used?
  • For the cell lines a short explanatory sentence about the cell type would be helpful e.g. for BUMPT  

Western Blot

  • Please insert in what the primary antibody was diluted and the secondary antibodies (+ dilution) in each case.
  • Please inserts the normalization between your band of interest and the housekeeping gene and show it in a bar graph and evaluate statistically. The authors stated that every western blot experiment have been repeated three times, so a quantification of all blots should be done.
  • The blots should not be cut because non-specific binding cannot be detected.
  • Figure 8B: Why is the actin band so weak?
  • Figure 8D and G: In the total blot, you can see clearly more bands than shown in the figure. How can this be explained and why is one sure that an equally intense band is the specific one?
  • What is the difference between 8D and E?
  • Figure 8F: the original data showed that the membrane was cutted in a lower and an upper part as before, but how can the 30/40 kDa marker band be on both halves of the same membrane? And if this is not the same membrane, why have the membrane always been cutted in such small pieces and how was an equal protein load/normalization to the house keeping protein done exactly?
  • Figure 8G: It is clear to see that the gang of housekeeping genes is not from the same blot therefore what is the reason for this and what conclusion can be drawn from this blot?

RNAseq Data:

·         Statistics are almost completely missing.

·         In the UMAP blots take always the same clusters and in the Dot blots look for the same genes in mouse and human.

  • Figure 1B-D: Why are the Violin pots between mFeature _RNA and nCount_RNA completely different and day 3 and 14 and why not in Percentace_mt?
  • Figure 1F: How do the AKI and Sham groups differ?
  • Figure 2C: But what about alcoholism and salmonella infection? These are also more active in failed repair PT subtypes.
  • If in Figure 2D the individual distinctions are merged again for simplicity, these main groups should be used everywhere, or in total the subgroups.
  • I do not see the identical signatures between Figure 4 J and K.
  • In Figure 4 the snRNA seq data were compared to published scRNA seq data. Is this reasonable?
  • What is the difference between Figure 5 C and D and Figure 5 E and F?
  • Figure 5E and F: Macrophages don't communicate with any other cell type? Please comment.
  • Figure 6 C and D: What are the arrows supposed to show me?
